# *Read-ME*: Refactorizing LLMs as Router-Decoupled Mixture of Experts with System Co-Design

**Ruisi Cai**[*1], **Yeonju Ro**[*1], **Geon-Woo Kim**[1], **Peihao Wang**[1],
**Babak Ehteshami Bejnordi**[2], **Aditya Akella**[1], **Zhangyang Wang**[1]
[1]The University of Texas at Austin, [2]Qualcomm AI Research
{ruisi.cai, gwkim, peihaowang, atlaswang}@utexas.edu,
{yro, akella}@cs.utexas.edu, behtesha@qti.qualcomm.com

## Abstract

The proliferation of large language models (LLMs) has led to the adoption of Mixture-of-Experts (MoE) architectures that dynamically leverage specialized subnetworks for improved efficiency and performance. Despite their benefits, MoE models face significant challenges during inference, including inefficient memory management and suboptimal batching, due to misaligned design choices between the model architecture and the system policies. Furthermore, the conventional approach of training MoEs from scratch is increasingly prohibitive in terms of cost. In this paper, we propose a novel framework *Read-ME* that transforms pre-trained dense LLMs into smaller MoE models (in contrast to "upcycling" generalist MoEs), avoiding the high costs of ground-up training. Our approach employs activation sparsity to extract experts. To compose experts, we examine the widely-adopted layer-wise router design and show its redundancy, and thus we introduce the pre-gating router decoupled from the MoE backbone that facilitates system-friendly pre-computing and lookahead scheduling, enhancing expert-aware batching and caching. Our codesign therefore addresses critical gaps on both the algorithmic and system fronts, establishing a scalable and efficient alternative for LLM inference in resource-constrained settings. *Read-ME* outperforms other popular open-source dense models of similar scales, achieving improvements of up to $10.1\%$ on MMLU, and improving mean end-to-end latency up to $6.1\%$. Codes are available at: https://github.com/VITA-Group/READ-ME.

## 1 Introduction

The success of Mixture-of-Experts (MoE) [1, 2] - such as recently exemplified by the Mixtral model [3] in the era of large language models (LLMs) - lies in its remarkable ability to leverage the collective expertise of specialized sub-networks, or "experts," each proficient in handling specific subsets or aspects of the data. By dynamically routing data through these experts, MoE models effectively capture complex patterns, adapt to diverse data distributions, and offer superior predictive accuracy compared to traditional single-model approaches. In addition to performance promise, MoEs also have a natural appeal for resource-limited devices due to their high sparsity, and therefore reduced activated parameters per token, which can potentially save inference costs [4, 5, 6, 7].

However, MoE inference presents significant challenges for key system-level objectives:

- **Memory Management:** Although MoEs activate only a subnetwork during inference, expert selection is determined on the fly by a layerwise router, complicating efficient prefetching. This often forces reliance on naive prefetching algorithms. For example, prior work has

---

[*]Equal contribution: authors are listed alphabetically. A. Akella and Z. Wang also advised this work equally.

38th Conference on Neural Information Processing Systems (NeurIPS 2024).

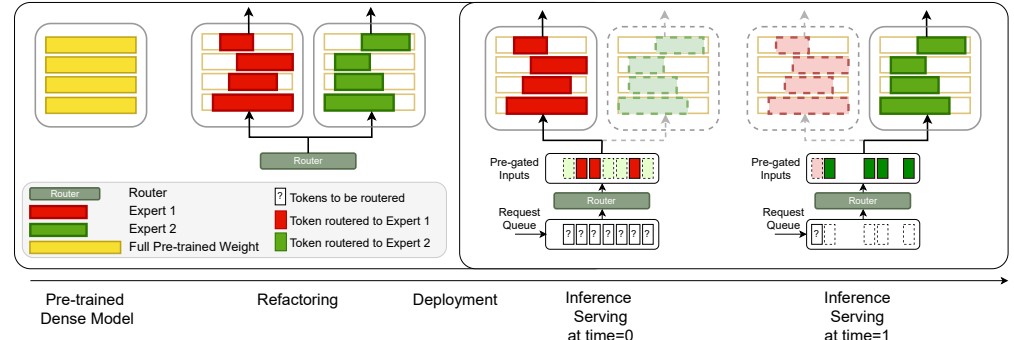

Figure 1: Overview of *Read-ME*. This figure shows the refactoring of a pre-trained dense model (in **yellow**) into two experts (in **red** and **green**). After refactoring, the model is deployed, and the serving timeline is depicted. At time $t = 0$, multiple inference requests (each a sequence of tokens) are queued, with expert assignment for each token undecided ("?") until processed by the router. Our router pre-gates tokens before inference, enabling expert-aware batching. Tokens are routed to their respective experts and batched accordingly: at $t = 0$ for Expert 1 (**red**) and at $t = 1$ for Expert 2 (**green**). New tokens enter the queue at each time step, with routing computed only for incoming tokens marked "?".

> predicted the next expert using hidden states from the previous layer and applied an LRU cache replacement for recently used experts [8]. While effective under certain conditions, such strategies depend on assumptions about expert locality and token predictability, which can become sub-optimal if those assumptions are violated (as shown in Table 4).
>
> - **Token Batching:** Token batching techniques critical for efficient inference (e.g., [9]) are poorly suited to MoE architectures, where each batch contains tokens invoking different experts across layers, rendering batching strategies ineffective (§ 4.2).

Moreover, traditional MoEs are typically trained from scratch, which becomes prohibitively expensive as model scales increase. To mitigate this, some approaches, such as "upcycling" [10], reuse pre-trained dense LLMs to initialize experts in an MoE. While that efficiently scales MoEs by leveraging smaller, pre-trained models, it does not address the inference-related challenges mentioned earlier.

In this work, we tackle the opposite challenge: *how to create a smaller MoE model from larger pre-trained models that enables resource-efficient inference while minimizing training costs?* Despite existing efforts [11, 12, 13, 14], this problem remains underexplored. Approaches like [11, 12, 13] attempt MoE refactorization but still adopt systems-unfriendly layer-wise structures for inference. Similarly, [14] focuses on dynamically selecting "important" neurons during pre-filling and pruning others during generation, but this is limited to long-content generation and requires neuron importance identification for each sequence.

To address both training and inference challenges, we introduce a holistic MoE framework dubbed *Read-ME*. To minimize training costs, we "refactorize" a pre-trained dense LLM into specialized experts through activation sparsity and optimize the routing policy (§ 3). For efficient inference, we examine the redundancy of layer-wise routers (§ 2.1, § 2.2) and propose decoupling the router from the MoE backbone (§ 2.3). This allows us to *pre-gate all requests (token sequences) before inference* and apply lookahead scheduling based on the experts to which tokens will be dispatched. Consequently, we propose an expert-aware batching algorithm (§ 4.2) and an optimal expert caching strategy inspired by Belady's offline caching algorithm [15] (§ 4.1).

Figure 1 illustrates our framework. Our key contributions are:

- We transform large pre-trained LLMs into Mixture-of-Experts (MoE) models with fewer activated parameters and small additional training cost (1 billion tokens). Our approach outperforms popular open-source models and compression techniques of similar scale on downstream tasks like MMLU [16].
- We analyze the widely adopted layer-wise routers in existing MoEs and reveal design redundancies. Current caching policies and batching algorithms are poorly suited to layer-wise MoEs. We propose a novel pre-gating router, decoupled from the MoE backbone, enabling better system-level optimization.

- Our system achieves a 6.1% reduction in mean latency and a 10% improvement in tail latency compared to state-of-the-art systems. Our caching algorithm is both provably and empirically optimal, thanks to our algorithm-system co-design.

## 2 Pre-gating Sparse Mixture of Experts

In this section, we introduce our motivation and design of pre-gating MoE which enables system-level acceleration by sharing and precomputing expert selection for each layer.

### 2.1 System Drawbacks of Conventional Sparse MoE Design

An Mixture-of-Expert (MoE) [1, 2, 17, 3] layer consists of a routing network $G$ and a set of $N$ expert networks $\{F_1, ..., F_N\}$. In the forward pass, the routing network will first process input sequences and generate the gating weights. Then a size-$K$ subset of experts will be dynamically activated and their outputs will be combined as final outputs according to the gating weights. In LLMs, MoE is typically adopted in the Feed-Forward Networks (FFN) within each transformer block [1, 2, 3]. Suppose an LLM has $L$ layers, the output of the $l$-th layer can be formulated as:

$$\boldsymbol{y} = \sum_{i=1}^{N} \mathbb{I}(|\{j \in [N] : G^{(l)}(\boldsymbol{x})_j \geq G^{(l)}(\boldsymbol{x})_i\}| \leq K) G^{(l)}(\boldsymbol{x})_i F_i^{(l)}(\boldsymbol{x}), \tag{1}$$

where the superscripts indicate the layer indices, $G^{(l)}, F^{(l)}$ are point-wise functions operating on tokens individually, and $\mathbb{I}(\cdot)$ is the indicator function which filters experts with top-$K$ gating weights. For shorthand, we denote $\mathbb{I}_i^{(l)} = \mathbb{I}(|\{j \in [N] : G^{(l)}(\boldsymbol{x})_j \geq G^{(l)}(\boldsymbol{x})_i\}| \leq K)$.

As shown in Eq. 1, conventional MoEs assign a separate router to each layer. While this is commonly used by open-source MoEs like Mixtral [3] and OpenMoE [18], we highlight its system inefficiency. Layer-wise gating makes it difficult to predict which expert to load until runtime (§ 4.1), and complicating request batching (§ 4.2). Specifically, layer-wise routers select the $l$-th layer expert $i : \mathbb{I}_i^{(l)} = 1$ based on the $(l-1)$-th layer outputs, which prevents pre-scheduling and pre-loading of data or model weights. This issue is especially problematic for billion-level parameter MoEs, where experts are usually distributed across devices (GPUs and CPUs in a machine) or even machines; in such situations, layer-wise selection accentuates high overheads of data I/O and communication among servers in the critical path of inference.

### 2.2 Redundancy of Layer-wise Router

In this section, we demonstrate that layer-wise gating patterns are redundant in an MoE. In particular, we empirically find that expert selections between two adjacent layers are highly correlated.

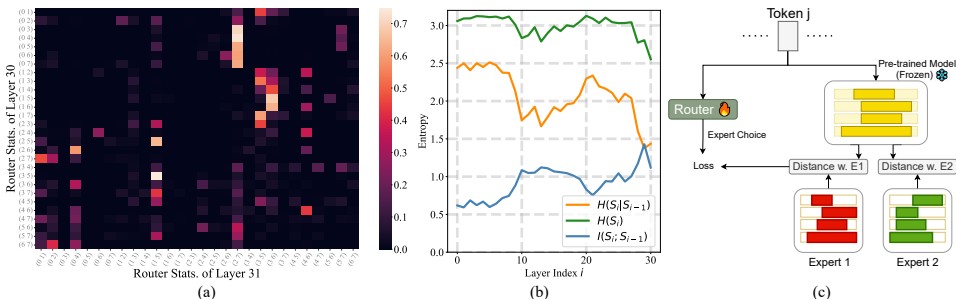

Figure 2: (a) Visualization of transition matrix between the $(l\text{-}1)$-th layer and the $l$-th layer, where each coordinate $[\{s, t\}, \{i, j\}]$ represents $P(\mathcal{S}^{(l)} = \{i, j\}|\mathcal{S}^{(l-1)} = \{s, t\})$. The row-wise sparse pattern suggests that the router decision becomes almost deterministic given the previous layer's decision. (b) Mutual information $I(\mathcal{S}^{(l)}; \mathcal{S}^{(l-1)})$, which indicates the learned knowledge shared by two neighboring layers is high. (c) Overview figure of router tuning and router distillation loss.

We use Mixtral-8×7B ($N = 8, K = 2$) [3] as a study case and analyze router decisions among its layers. Define the random variable $\mathcal{S}^{(l)} = \{i \in [N] : \mathbb{I}_i^{(l)} = 1\}$ as the pair of experts selected

for each layer ($|\mathcal{S}^{(l)}| = 2$). We are interested in the conditional probability of $\mathcal{S}^{(l)}$ between two consecutive layers: $P(\mathcal{S}^{(l)} = \{i, j\}|\mathcal{S}^{(l-1)} = \{s, t\})$. The transition matrix of the last two layers from Mixtral-8×7B is depicted in Figure 2 (a). The row-wise sparse pattern implies that the expert selection is almost deterministic given the previous layer' choices. For example, for tokens choosing expert-3 and expert-5 in the 30th layer, over 70% will select expert-1 and expert-5 at the 31st layer.

To further validate this observation, we plot the mutual information between expert choices of every two neighboring layers: $I(\mathcal{S}^{(l)}; \mathcal{S}^{(l-1)})$. As reflected in the right sub-figure of Figure 2, knowing expert pairs used in the last layer significantly reduces the uncertainty of the next layer. Thus, the implicit knowledge learned by each router is extensively shared across layers.

### 2.3 Pre-Computed Routing Policy

The above observations suggest that among the many $\binom{N}{K}^L$ routing paths, only a few are used during the inference. Therefore, layer-wise routing decisions are unnecessary for MoEs. Instead, we can separate the routerfrom the MoE backbone and pre-compute the routing path all at once.

First of all, we assume the indices of experts handling one domain of tokens are aligned, i.e. $\{F_i^{(1)}, \cdots, F_i^{(l)}\}$ always forms a routing path. We defer our approach to the construction of aligned experts to §3. Next, we let a singleton network $G$ generate gating weights for all layers. In particular, we adopt one transformer block with causal attention as the model architecture of $G$. Gating weights computed in this way not only leverage the states of the current token but also take the information from the past tokens into consideration. Thus, tokens will have expert selections similar to the recent tokens, which ensures cache-friendly inference (see more details in § 5.3).

Suppose the input sequence is $(\boldsymbol{x}_t)_{t=1,\cdots,T}$, the output for the $t$-th token at the $l$-th layer is:

$$\boldsymbol{y}_t = \sum_{i=1}^{N} \mathbb{I}(|\{j \in [N] : G(\boldsymbol{x}_{\leq t})_j \geq G(\boldsymbol{x}_{\leq t})_i\}| \leq K) G(\boldsymbol{x}_{\leq t})_i F_i^{(l)}(\boldsymbol{x}_t), \qquad (2)$$

where $\boldsymbol{x}_{\leq t} = (\boldsymbol{x}_1, \cdots, \boldsymbol{x}_t)$ represents all the tokens before the $t$-th token. We note that $G$ is independent of layer index $l$. Despite a subtle change, it brings profound benefits to enable system-level optimization. In brief, by separating the gating network from the transformer layers, expert selection can be determined at the outset and used to schedule the data-loading procedure for each layer. We defer more details on system co-design to §4.

## 3 Re-factoring Language Model with Pre-Gating MoE

In this section, we introduce the main technique to re-use a dense pre-trained model to construct our pre-gating MoE proposed in §2. In short, our approach first initializes each expert by structured pruning of a dense model on the corresponding data domains. Afterward, we instantiate a gating network shared across layers and continue joint training of the router and experts.

**Domain-Aware Expert Construction.** We construct a set of small experts by pruning the dense model with different data domains. To begin with, we point out that public language corpora often contain metadata indicating the domain of each subset. For example, the training dataset of LLaMA family [19] can be split into scientific articles [20], novels [21], and QAs [22], etc. We utilize this metadata to group data entries in the training corpus into $N$ sub-domains $\{\mathcal{D}_1, \cdots, \mathcal{D}_N\}$. Observing that feature channels on each subset are sparsely activated [23], we compute the average magnitude of a channel on each subset and keep top activated neurons to form the domain expert. Formally, let the number of experts equal to the number of sub-domains, and assume the dense model is a two-layer FFN with hidden size $D$: $F_0(\boldsymbol{x}) = \boldsymbol{W}_2\sigma(\boldsymbol{W}_1\boldsymbol{x})$, then the $i$-th experts with hidden size $d$ are initialized as: $F_i(\boldsymbol{x}) = \boldsymbol{W}_2\boldsymbol{M}_i^\top\sigma(\boldsymbol{M}_i\boldsymbol{W}_1\boldsymbol{x}), \forall i \in [N]$, in which $\boldsymbol{M}_i$ is obtained by:

$$\underset{\boldsymbol{M}\in\{0,1\}^{d\times D}}{\arg\max} \ \mathbb{E}_{\boldsymbol{x}\sim\mathcal{D}_i}\|\boldsymbol{M}\boldsymbol{W}_1\boldsymbol{x}\|_1 \quad \text{s.t.} \quad \boldsymbol{M}\mathbf{1}_D = \mathbf{1}, \boldsymbol{M}^\top\mathbf{1}_d \leq \mathbf{1}, \qquad (3)$$

where $\boldsymbol{M}$ is constrained to be a selection matrix without replacement. The mask for each layer is jointly optimized so that the resultant experts are aligned layerwise and dedicated to the same data distribution. In our experiments, we set $d \approx D/2$. In addition, we observe that a certain

subset of channels is essential for all data, potentially due to the system prompt and the presence of commonsense knowledge. Therefore, we isolate the corresponding neurons as the *permanent expert*, which will be activated for all tokens, similar to previous designs [18, 24].

**Continual Training Objective.** After initializing experts via structured pruning, we perform joint training of randomly initialized gating networks and expert subnetworks via causal language modeling. In addition, we propose *routing distillation loss* to enhance the alignment between expert choice in pre-gating MoE and the activation sparsity in the original dense model.

We illustrate the training of our router in Fig. 2 (c). Suppose the predicted token has embedding $\boldsymbol{x}_{t+1}$. We feed $\boldsymbol{x}_{t+1}$ into the original dense model $F_0$ and get a sparse selection matrix $\boldsymbol{M}_0$ that indicates neurons with top 50% magnitude similar to Eq. 3. Then we penalize this loss function:

$$\mathcal{L}_{RD} = \mathcal{D}_{KL}\left(\mathrm{softmax}(G(\boldsymbol{x}_{\leq t+1}))\|\mathrm{softmax}([\|\boldsymbol{M}_0\boldsymbol{M}_1^\top\|_F^2, \cdots, \|\boldsymbol{M}_0\boldsymbol{M}_N^\top\|_F^2])\right). \quad (4)$$

Here, $\mathcal{D}_{KL}(\cdot\|\cdot)$ represents Kullback–Leibler divergence. $\|(\boldsymbol{M}_0\boldsymbol{M}_j^\top\|_F^2 = \mathbf{1}_d^\top \boldsymbol{M}_0\boldsymbol{M}_j^\top \mathbf{1}_d$ computes the Hamming distance between two masks induced by $\boldsymbol{M}_0, \boldsymbol{M}_j$. We apply softmax to normalize these scores as the estimated selection probability of each expert for predicted token $\boldsymbol{x}_{t+1}$.

## 4  Expert-aware Inference System

We demonstrate how our refactoring and pregating concepts enable a novel, high-performance, and efficient MoE inference method. We address two key challenges in existing MoE models' inference: inadequate memory management and limited support for batched inference. Our problem setting is broad, aiming to serve multiple requests using an MoE model, each comprising a sequence of tokens. This differs from previous systems, which focused on optimizing performance for individual requests.

### 4.1  Pre-gating Optimized Expert Prefetching and Caching

MoE models promise reduced memory usage during inference by loading only the parameters of required experts, skipping the rest. However, traditional layer-wise gating imposes significant loading costs. Previous approaches, such as on-demand loading [25], prefetching [26], and expert caching [8, 27], attempt to address this. However, on-demand loading adds overhead to the critical inference path, and prefetching often loads unnecessary experts due to incomplete routing information, leading to suboptimal memory usage and performance [28]. Additionally, caching strategies, based on request characteristics like temporal locality or activation sparsity, have mostly been evaluated in isolated single-request scenarios. In practice, expert caches are shared across multiple requests, making cache policies relying on per-request traits suboptimal. A global view across all requests is necessary for effective caching (see Table 4). Our work leverages pre-gating to develop more informed prefetching and caching strategies, resulting in significant system-level improvements.

**Fine-grained Prefetching.** By design, our pre-gating MoE architecture enables us to prefetch the exact expert layers needed for a token or a request, avoiding guesswork. To further hide the latency in prefetching, we pipeline and thus overlap loading of experts and experts' computation at layer-wise granularity: specifically, while computing the $i$th layer's forward path in the compute stream, we load the $i + 1$st layer's experts in a separate loading stream.

**Belady-inspired Caching.** Prefetching can hide the loading latency of all but the first layer, which incurs significant cost. To mitigate this, we need a cache that stores relevant initial layers, and we argue that pre-gating enables an optimal caching strategy.

The classical Belady algorithm is known to be the *optimal offline cache replacement algorithm*, replacing the object that will be accessed farthest in the future. While impractical in real-world systems (due to unknown future accesses), our pre-gating architecture allows us to approximate it. By decoupling the router from the backbone MoE, we can compute future expert references across requests in advance, enabling near-optimal cache replacement.

Suppose that the cache at time step $t - 1$ is as follows: $C(t - 1) = \{e_1, e_2, ..., e_k\}$, where the cache is of size $k$ and is filled with $k$ experts $e_{1...k}$. $F(e, t)$ represents the next time after $t$ when expert $e$ will be requested. Then, our policy chooses the expert $e_{evict} = argmax_{e \in C(t-1)} F(e, t)$ for eviction.

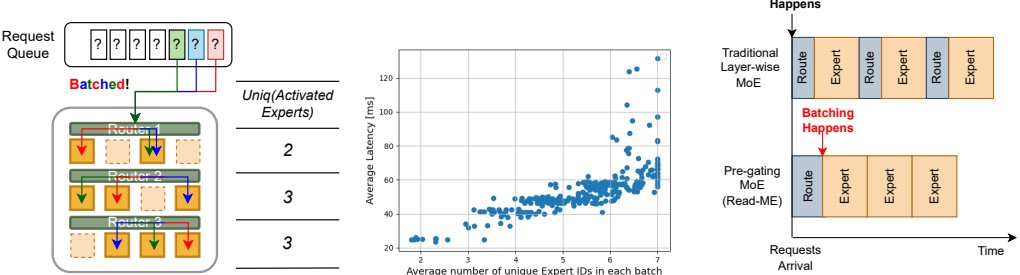

Figure 3: Challenges of MoE serving in current serving systems and *Read-ME*'s batching pipeline.

## 4.2 Expert-aware Batching

Current serving systems heavily rely on batching to improve inference efficiency, but effective batching for MoE models remains challenging. As shown in Figure 3 (a), each token in MoE models may invoke a different set of experts per layer, leading to multiple expert activations for a batch of requests. For example, in a toy model with 4 experts per layer and a batch of 3 tokens (one per request), 2/3/3 experts would be activated across the layers. In the Mixtral8x7B model [3] applied to the chatbot arena dataset [29], we observed an average activation of **7.63 out of 8 experts**, even with a modest batch size of 56.8.

The core challenge is that while each token requires computation from only one expert per layer, it must wait for all other tokens in the batch to complete their expert computations in the same layer [30]. This bottleneck repeats at each layer, reducing the efficiency of batching. Ideally, a single loaded expert would serve multiple tokens in a batch, but this is rarely achieved, affecting both performance and efficiency. For example, we observe a linear increase in average per-token processing latency as the number of unique experts per batch grows (see Figure 3 (b)).

In contrast, pre-gating enhances inference performance by enabling the *delayed creation of an optimal batch based on required experts*. For a given set of tokens, we pre-gate each one and select a subset for batching, depending on their identified expert requirements. The goal is to minimize the number of unique experts across all layers while maximizing the number of tokens in the batch. Moreover, as discussed in § 2.3, our expert selection remains consistent across layers—if a token is assigned to Expert 1, it will be routed to Expert 1 in every layer. This approach, combined with our batching strategy, ensures optimal efficiency. Algorithm 1 provides our batching pseudocode.

We note that in other MoEs, such batching isn't feasible because, as shown in Figure 3, their expert selection at each layer remains unknown until the request reaches the router. In *Read-ME*, experts are determined first, which allows batches to be created and submitted to MoE layers efficiently.

## 5 Evaluation

In this section, we start by describing the experimental details in § 5.1. Then we validate the refactorization effectiveness on downstream tasks in § 5.2. In § 5.3, we evaluate the effectiveness of pre-gating and batching. § 5.4 analyzes memory optimization techniques. In addition, we provide experimental details in § 5.1, and more experimental results in §. A.

Table 1: Details of router design. Following the standard Transformer architecture [31], the inserted router adds only 18 million additional parameters.

| | |
|---|---|
| # Layers | 1 |
| # Heads | 4 |
| Vocab size | 32000 |
| Embedding Dim. | 512 |
| Feature Dim. | 512 |
| MLP Intermediate Dim. | 512 |
| Activation Function | SwiGLU [32] |
| Positional Embedding | RoPE [33] |
| Normalization | RMSNorm [34] |
| # Params | 18.0 M |

### 5.1 Experimental Details

**Model and Dataset** We perform the MoE refactorization based on Llama2-7B-chat [19] model, a popular open-source model pre-trained on 2 trillion tokens. The training corpus [35] involves the data collected from 7 different resources: Arxiv [20], Books [21], Common Crawl, C4 [36], Github, Wikipedia [37], and StackExchange [22]. To generate experts, we collect 16 samples from each data domain, with each sample consisting of 4096 consecutive tokens. During router tuning, we use the subset of RedPajama dataet [35], with the same curation strategy. We present our detailed router design in Table 1. We use the standard Transformer [31] architecture with a 1-layer, 4-head

**Algorithm 1** `Read-ME` Expert-aware Batching Algorithm (pseudocode)
_____________________________________________________________________

    **Input** NumExperts, ReqQueueByExpert, MaxTokenLen
    **Output** ScheduledReq

1: **for** $k \leftarrow 0$ to $NumExperts - 1$ **do**
2:     $len\_reqs\_per\_experts[k] \leftarrow len(ReqQueByExpert[k])$
3: **end for**
4: **while** true **do**
5:     $E \leftarrow argmax(len\_reqs\_per\_experts)$
6:     **if** $len\_reqs\_per\_experts[E] < (MaxTokenLen - len(ScheduledReq))$ **then**
7:         $ScheduledReq \leftarrow ScheduledReq \cup ReqQueueByExpert[k]$
8:         $ReqQueueByExpert[E] \leftarrow [\ ]$
9:         $len\_reqs\_per\_experts[E] \leftarrow 0$
10:     **else if** $MaxTokenLen - len(ScheduledReq) \geq 0$ **then**
11:         $n\_available \leftarrow MaxTokenLen - len(ScheduledReq)$
12:         $ScheduledReq \leftarrow ScheduledReq \cup ReqQueueByExpert[k][: n\_available]$
13:         $ReqQueueByExpert[E] \leftarrow ReqQueueByExpert[k][n\_available :]$
14:         $len\_reqs\_per\_experts[E] \leftarrow len(ReqQueueByExpert[E])$
15:         break
16:     **else**
17:         break
18:     **end if**
19: **end while**
_____________________________________________________________________

design. The router is lightweight, consisting of 18 million additional parameters, and incurs negligible computational overhead. We use 8 A100 GPUs with 80GB of memory for all tuning experiments.

**Continual-Tuning Details** To co-optimize the router and expert networks, we iteratively tune each model component. Specifically, we first optimized the router by $\mathcal{L}_{RD}$, as detailed in § 3, for 100 steps. We use the batch size of 64 in this router tuning stage. During this *router tuning* stage, we freeze the expert weights and solely tune the router weights. Then, during the *expert tuning* stage, we fix the router weights and modify the expert weights via language modeling loss, for 200 steps, with a batch size of 128. We set sequence length to 4096 for all stages, following the choice in the pre-training stage of

Table 2: Hyper-parameter choice during the training.

| Stage | Router Tuning | Expert Tuning |
|---|---|---|
| # Iteration per Round | 100 | 200 |
| # Rounds | 8 | 8 |
| Initial LR at Round 0 | $5e^{-4}$ | $5e^{-5}$ |
| LR Decay within Round | Cosine | Cosine |
| LR Decay type across Rounds | Exponential | Exponential |
| LR Decay rate across Rounds | 0.8 | 0.8 |
| Weight Decay | 0.01 | 0.01 |
| Batch Size | 64 | 128 |
| Sequence Length | 4096 | 4096 |
| # Tokens per Round | 26 M | 105 M |
| # Tokens in Total | | 1.04 B |

Llama2 model [19]. This iterative training schedule is conducted 8 times. Detailed visualizations of the training dynamics are provided in Section A.1. For each round, the router tuning and expert tuning stages will cost 26 million and 105 million tokens, respectively. The whole continual-tuning process merely uses 1.04 billion tokens, negligible compared to the pre-training cost (2 trillion tokens). During each round of tuning, we use the cosine learning rate decay. At round 0, the initial learning rates are $5e^{-4}$ for router tuning and $5e^{-5}$ for expert tuning. The initial learning rate decays exponentially with a decay rate of 0.8 as the number of rounds increases.

**Inference System Evaluation** For our workload, we utilize the Chatbot Arena Conversation Dataset [29] to generate inference requests and replay conversation traces. Our setup employs a single A100 GPU with 80GB of memory. The implementation is built on top of DeepSpeed inference engine [38]. We use normalized latency as our primary metric, defined as the end-to-end latency divided by the generated token length, in line with previous works [9, 39, 38].

## 5.2 Downstream Task Evaluations

We first validate the refactorization effectiveness on downstream tasks, as shown in Table 3, comparing it to other models of similar scales, including the open-source models that trained from scratch, and

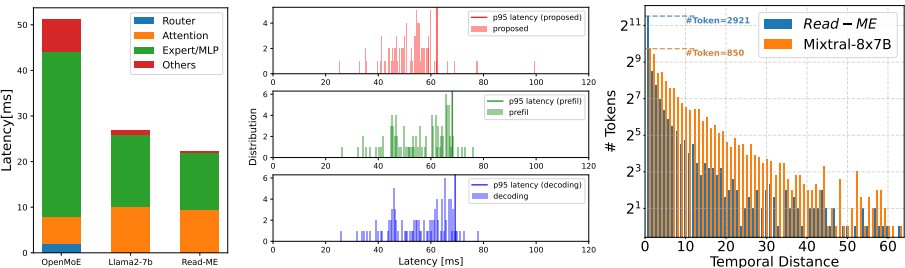

Figure 5: Latency evaluation and Temporal locality analysis. (Left) Single inference latency measured on a 124 token generation task. (Center) Latency distribution measured on synthetic workload replaying Chatbot Arena Dataset [29] (§ 5.1). (Right) Temporal distance measured on Arxiv dataset [20], and a subset of Redpajama [35].

the dense models pruned from larger pre-trained LLMs. We achieve the best average performance, outperforming all model variants from the Pythia [40] and Open-Llama-v2 [41] families, as well as Sheared-Llama [42]. We use just 1 billion training tokens, considerably less than other models.

Table 3: Downstream task evaluation of our proposed method (*Read-ME*) compared to open-source models, including dense models Pythia and Open-Llama-v2, the MoE model OpenMoE, and the compression method Sheared-Llama. The evaluation includes zero-shot performance on WinoGrande, ARC-Easy, LogiQA, CoQA; 5-shot performance on MMLU; 10-shot on Hellaswag; and 25-shot on ARC-Challenge. The "#Param" column presents in the form of (# Activated-Parameters - # Total-Parameters). Training cost is measured by the number of tokens used. For compression methods like ours and Sheared-Llama, only tokens used for conversion are counted, excluding Llama-2 pre-training costs.

| Method | #Param | Cost | MMLU | Hell. | Wino. | ARC-E | ARC-C | LogiQA | CoQA | avg. |
|---|---|---|---|---|---|---|---|---|---|---|
| Sheared-Llama | 2.7B | 50B | 26.4% | **70.8**% | 67.0% | **67.0**% | 41.2% | 28.3% | 71.7% | 53.2% |
| Pythia | 2.8B | 300B | 26.9% | 60.8% | 59.7% | 64.4% | 36.4% | 27.7% | 61.9% | 48.3% |
| Open-Llama-v2 | 3.4B | 1T | 25.7% | 67.6% | 63.5% | 66.5% | 39.0% | 28.1% | 54.4% | 49.3% |
| OpenMoE | 2.1B-8B | 1.1T | 26.2% | 45.5% | 60.3% | 64.1% | 30.3% | - | - | - |
| *Read-ME* | 4.7B-17B | 1B | **38.9**% | 68.5% | **67.7**% | 66.6% | **42.3**% | **29.7**% | **74.8**% | **55.5**% |
| Pythia | 6.9B | 300B | 25.5% | 67.1% | 64.1% | 67.3% | 31.3% | 25.3% | 63.6% | 49.2% |
| Open-Llama-v2 | 6.9B | 1T | 40.2% | 66.7% | 66.0% | 63.0% | 36.0% | 27.6% | 64.5% | 52.0% |
| Llama-2 | 6.9B | 2T | 45.3% | 78.6% | 69.3% | 76.4% | 53.0% | 31.0% | 75.9% | 61.4% |

In Fig. 4, we further provide a direct comparison with other compression methods, which converts a large LLM to a small dense variant, on MMLU [16] benchmarks. Besides open-source models and Sheared-Llama [42] which are mentioned in the previous table, we additionally include recent compression techniques, including LLM-Pruner [43], SliceGPT [44], LaCo [45], and Compresso [46], as our baselines. *Read-ME* achieves the best performance among the models with the number of activation parameters less than 5 billion, and shows comparable performance with Open-Llama-v2-7B [41]. More analysis is included in § A.2.

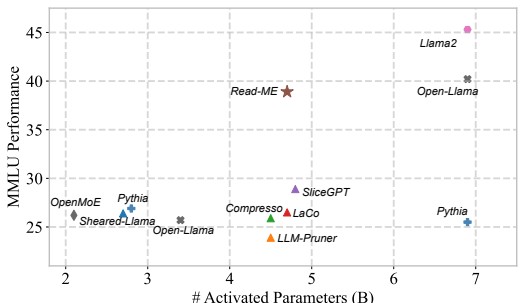

Figure 4: Evaluation of *Read-ME* on MMLU [16] benchmark, compared to other open-source models and compression techniques ( performance numbers are collected from their respective papers.

## 5.3 Pre-gating and Expert-aware Batching

**Inference Latency Breakdown.** We evaluate the impact of the auto-regressive router introduced by our refactoring of the dense MoE on per-request inference latency. Unlike conventional layer-wise routers, usually linear layers, our auto-regressive router comprises a multi-head attention layer and an MLP layer (see § 2.3), potentially raising its computational cost.

Fig. 5 (left) illustrates the average per-token latency breakdown of a single isolated inference request measured in `OpenMoE` [18] with conventional layerwise routers, our refactored model with pregating router, and the original dense Llama2-7b model [19] we refactored. We find that the computational

overhead of our auto-regressive router is minimal – its contribution of 0.4% is much less compared to the router's net contribution in other MoE models (3.95%). This is because we use a single router unlike other models with gating for each MoE layer; also, our router design is compact with only 18M parameters (Table 1). Compared to the dense model, we achieve a net 19% reduction in latency via refactoring the MLP to MoE.

**Batched Inference.** We now evaluate the efficacy of our expert-aware batching. Fig 5 (center) shows the latency distribution and the 95-th percentile latency (p95) during batched inference. We compare with two widely used techniques – Decoding-prioritized batching [38], and Prefill-prioritized batching [39, 47]. These methods utilize distinct queues for decoding requests and prefill requests, prioritizing batching of tokens from decoding and prefill requests, respectively.

Prioritizing either decoding or prefill requests yields comparable performance. In contrast, our method of constructing batches based on activated experts enhances the mean latency by 5.0-6.1% and reduces the p95 latency by 9.5-10.0% compared to these approaches.

The primary reason for this improvement is that our batching approach directly reduces the average number of unique experts invoked per batch by leveraging pre-gated information. Specifically, for decoding-prioritized and prefill-prioritized batching, the average number of unique experts per batch was 5.08 and 5.21, respectively, whereas our method reduces this to 3.51.

We observed a significant performance impact as prefill requests invoke more experts per batch compared to decoding requests. Prefill requests require tokens to be dispatched to different experts, making it impractical to batch tokens by shared experts due to attention operations. As a result, a substantial number of experts are invoked for each batch, negatively affecting performance. Fortunately, our auto-regressive router design improves temporal locality in prefill requests, often allowing tokens within the same request to select the same or a small number of experts. We explore this locality in greater detail in the following section.

**High Temporal Locality.** To analyze the locality, we measure the temporal distance of the tokens in a sequence (Fig. 5 (c)). We define temporal distance as the distance between two tokens selecting the same expert within a sequence [48]. Our result shows that our router leads to a smaller distance, indicating a high degree of temporal locality. Specifically, out of 4096 tokens, 2921 tokens follow the choice of the last token, compared to 850 tokens in Mixtral-8×7B. The locality is attributed to the auto-regressive design of our router, where the router's decision is based on the current and all previous tokens. As a result, a given token is likely to have similar expert selections with its recent predecessor tokens. However, note that this temporal locality appears only within the token sequence of a single request and does not appear across different requests.

## 5.4 Memory-Efficient Inference

We evaluate how well our approach can ensure good performance while improving memory efficiency. In particular, we constrain the expert cache capacity to $k$ (that is, up to $k$ experts can reside in accelerator memory). In this setup, if a requested expert is not in memory, it must be loaded from host memory, potentially increasing loading latency. As explained in § 4.1, this loading overhead can be mitigated with prefetching, provided that we know which expert will be needed in *Read-ME*. We compare the end-to-end latency of requests from the prefetching our approach enables (`Prefetching`) versus not leveraging prefetching (`On-demand Loading`) [25]. Figure 6 shows that for varying cache capacities, we consistently outperform `On-demand Loading`, with up to 30% better latency.

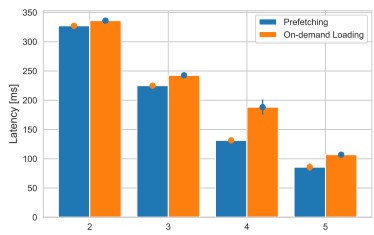

Figure 6: Latency impact of prefetching: We measured end-to-end latency on a synthetic workload generated by replaying Chatbot Arena Dataset [29]. (Appendix 5.1)

In addition to proactively loading experts into memory, our approach also retains experts in a cache to further use memory optimally. Table 4 compares three representative caching policies' hit ratios across varying cache capacities, including the Belady-inspired approach that our architecture enables. As noted earlier, our approach accommodates multiple requests where each request has a token sequence, in contrast with prior works focusing on a single request/token-sequence [8, 27].

When multiple requests share the expert cache, temporal locality within a single request cannot be leveraged across requests, limiting its effectiveness. This explains why LRU, which works well in single-request scenarios, underperforms in our setup. In contrast, our Belady-based algorithm excels at all cache capacities by utilizing future expert information across requests, thanks to the pre-gating router. When cache capacity is constrained by system memory, latency can be significantly reduced with an optimized cache policy. Our Belady approach notably improves latency, particularly under limited cache sizes, though we omit detailed results for brevity.

| Cache Capacity | Cache Policy | | |
| --- | --- | --- | --- |
| | Random | LRU | Belady |
| 2 | 34.19% | 33.90% | **44.16%** |
| 3 | 50.14% | 52.42% | **61.82%** |
| 4 | 67.52% | 66.95% | **77.21%** |
| 5 | 82.91% | 83.48% | **88.03%** |

Table 4: Cache hit ratio measured in batched inference setup.

## 6 Related Work

**MoE Refactorization.** Recent "MoE-fication" methods [11, 12, 13, 49] optimize or group channels using graph-based techniques but still rely on system-inefficient layer-wise routers. In contrast, we are the first to identify the redundancy in layer-wise routers and propose a pre-gating router that enables expert pre-fetching. Similar to [50, 14, 51], we leverage activation sparsity [23] to construct experts, adaptively identifying important neurons and evicting less-important ones during inference.

**Efficient Inference Serving.** To deal with the limited memory in resource-constrained settings, prior LLM inference works focused on optimizations such as offloading parameters to host memory [52, 53, 25], quantization [54, 55, 56], sparsity [57, 58] and MoE architectures [4, 59, 26]. However, while token batching [9] has garnered significant attention for dense models [39, 47, 38, 60], it remains problematic and underexplored in the context of MoE models.

Pre-gated MoE [28] is related to *Read-ME* as they too fine-tune a router to pre-gate using the $i$th layer's hidden states to compute the $i + 1$th layer's routing; but they still maintain a layer-wise architecture which constrains batching. SiDA-MoE [61] separates the router from the inference path. However, tokens cannot be batched together because they do not share routing decisions across all layers. In addition, the offline routing function of SiDA is an approximation that may incorrectly guess expert selection, especially when the model scales. In contrast, *Read-ME* has exact routing, ensuring no performance drop during inference.

Mixtral-offloading [8] introduces speculation to "guess" routing decisions, resorting to costly on-demand loading if speculation fails. Caching is commonly used [62, 52, 63, 53, 64], including in MoE systems [8, 27], which typically focus on single requests. Prior caching methods are limited by layer-wise routing and lack of foresight into future requests.

## 7 Conclusions and Limitations

We address the under-explored challenge of reusing a pre-trained LLM to create a smaller MoE model that enables efficient inference with minimal training cost. By leveraging activation sparsity, we construct specialized experts and integrate them via a router. Upon analyzing the layer-wise router design used in all open-source MoEs, we identify its inefficiency and redundancy. To overcome this, we propose a pre-gating router, decoupled from the MoE backbone, enabling system-level optimizations that were previously unattainable.

**Limitations.** Our serving system is designed for a single accelerator, and extending it to distributed serving remains a non-trivial task for future work. Our method has no negative societal impact, as it uses publicly released data and model checkpoints. This work is foundational research and is not tied to specific applications.

## Acknowledgements

The work of Z. Wang is in part supported by the US Army Research Office Young Investigator Award (W911NF2010240) and a Research Gift from Qualcomm. Ro and Akella are supported by NSF grants CNS-2105890 and CNS-2232135 and by Cisco Research and Meta.

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

## A  More Experimental Results

### A.1  Training Dynamics

As detailed in § 5.1, we iteratively tune the router and experts for 8 rounds. We visualize the validation loss during the first 4 rounds out of the total 8 rounds of training. In Fig. 7, the router tuning stages are marked in gray, while the expert tuning stages are marked in orange. Two observations can be drawn from Figure 7: (1) The validation loss decreases during both router tuning and expert tuning stages. (2) The validation loss reduction from router tuning saturates after two rounds, while the validation loss continues to decrease during expert tuning.

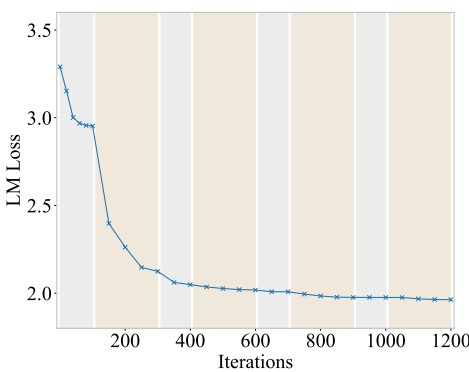

Figure 7: Visualization on training dynamics.

### A.2  MoE Achieves Better Efficiency-Accuracy Trade-off than Dense Models.

Prior compression-based works [42, 43, 44, 45, 46] focus on converting a large *dense* pre-trained model into a smaller *dense* model. However, we argue that a smaller *MoE* model (i.e. the MoE model with the smaller number of activation parameters) is a better target architecture. To ensure a fair comparison, we (1) derive a small dense model with $4.7B$ parameters, matching the size of a single expert network, using the same amount of data, and (2) fine-tune the obtained dense model for an equivalent number of steps. As shown in Table 5, refactorizing the pre-trained model into an MoE structure, rather than a smaller dense variant, leads to significant performance improvement. The models are evaluated based on performance on the MMLU [16], and perplexity across seven data domains included in RedPajama [35].

Table 5: We compare the *Read-ME* performance with dense model, and report the MMLU performance and perplexity on 7 data domains. By adopting an MoE as the target structure instead of dense model, our model achieve significantly better overall performance.

| Evaluation | Arxiv | Books | C4 | Common Crawl | Github | StackExchange | Wikipedia | MMLU |
|---|---|---|---|---|---|---|---|---|
| Dense | 5.63 | 1.94 | 11.78 | 9.68 | 3.75 | 13.42 | 6.24 | 27.1% |
| *Read-ME* | 4.18 | 1.31 | 10.57 | 7.72 | 2.39 | 12.52 | 3.94 | 38.9% |

### A.3  *Read-ME* Remains Effective without Prior Knowledge of the Training Domain

We additionally use the Mistral [65] model as the pre-trained dense model, and convert it to the MoE structure, with the proposed method. The task is challenging because we do not have prior knowledge on the Mistral original training data, and our experiment in Table 6 shows that our method remains effective without the prior knowledge of the original training domain.

Table 6: Ablation study on Mistral [65] pre-trained model.

| Method | Pre-trained Domain | Fine-tune Domain | #Param | MMLU | Hell. | Wino. | ARC-E | ARC-C | LogiQA | CoQA | avg. |
|---|---|---|---|---|---|---|---|---|---|---|---|
| *Read-ME*-Llama-2 | Red-pajama | Red-pajama | 4.7B-17B | 38.9% | 68.5% | 67.7% | 66.6% | 42.3% | 29.7% | 74.8% | 55.5% |
| Llama-2 | Red-pajama | - | 6.9B | 45.3% | 78.6% | 69.3% | 76.4% | 53.0% | 31.0% | 75.9% | 61.4% |
| *Read-ME*-Mistral | N/A | Red-pajama | 4.7B-17B | 39.2% | 79.1% | 68.2% | 77.1% | 49.3% | 30.9% | 76.2% | 60.0% |
| Mistral | N/A | - | 6.9B | 62.1% | 84.5% | 79.3% | 82.7% | 63.7% | 33.5% | 80.3% | 69.4% |

### A.4  Computational Cost of Auto-regressive Router

For a detailed cost analysis of auto-regressive router that we introduced, we added: (1) FLOPs comparison, (2) latency, and (3) latency breakdown with a larger batch size (high-throughput scenarios) of a Traditional Router (TR) and an Autoregressive Router (AR). To focus solely on the router's impact on latency, we controlled other variables (e.g., the number of activated parameters) to be the same.

Note that the computational cost of both the traditional router and the autoregressive router is theoretically *linear to batch size*. Therefore, when the batch size is high (in high-throughput

Table 7: Flops comparison between Traditional router and Auto-regressive router

|  | Traditional Router | Auto-regressive Router |
|---|---|---|
| Flops/sample | 4.7 KFLOPs | 3 KFLOPs |

Table 8: Latency [ms] comparison between Traditional router and Auto-regressive router

|  | bsz=5 TR | bsz=5 AR | bsz=10 TR | bsz=10 AR | bsz=20 TR | bsz=20 AR | bsz=30 TR | bsz=30 AR |
|---|---|---|---|---|---|---|---|---|
| Router | 1.76 | 0.61 | 1.80 | 0.61 | 1.78 | 0.61 | 1.93 | 0.61 |
| Attention | 18.13 | 18.18 | 18.28 | 18.13 | 18.49 | 18.36 | 19.59 | 19.66 |
| Expert/MLP | 22.43 | 21.75 | 24.59 | 22.53 | 24.97 | 22.99 | 30.17 | 28.31 |
| Sum | 42.31 | 40.55 | 44.67 | 41.27 | 45.23 | 41.96 | 51.69 | 48.59 |

Table 9: Latency breakdown comparison between Traditional router and Auto-regressive router

|  | bsz=5 TR | bsz=5 AR | bsz=10 TR | bsz=10 AR | bsz=20 TR | bsz=20 AR | bsz=30 TR | bsz=30 AR |
|---|---|---|---|---|---|---|---|---|
| Router | 4.15% | 1.50% | 4.02% | 1.48% | 3.93% | 1.46% | 3.74% | 1.26% |
| Attention | 42.85% | 44.85% | 40.92% | 43.92% | 40.87% | 43.75% | 37.90% | 40.47% |
| Expert/MLP | 53.01% | 53.65% | 55.06% | 54.59% | 55.20% | 54.80% | 58.36% | 58.27% |
| Sum | 100.00% | 100.00% | 100.00% | 100.00% | 100.00% | 100.00% | 100.00% | 100.00% |

scenarios), the cost increases linearly. In both cases, the computation can be parallelized, so this remains negligible in end-to-end latency even in high-throughput scenarios. In fact, we would like to clarify that the bottleneck in high-throughput scenarios is actually the expert layers, as seen in Table 9 – Expert/MLP row. This issue can be addressed by the methods discussed in Section 4. *Traditional layerwise routers do not allow for efficient system design*, which underscores the need for a careful co-design of routers.

