# OpenReview forum: "$\textit{Read-ME}$: Refactorizing LLMs as Router-Decoupled Mixture of Experts with System Co-Design"
_NeurIPS.cc/2024/Conference — NeurIPS 2024 poster_

### Official Review · Reviewer_AX77 · 2024-07-04

**Soundness:** 3
**Presentation:** 3
**Contribution:** 2
**Rating:** 4
**Confidence:** 4

**Summary:**

This paper proposes to perform data-specific pruning from a regular LLM and, with a small amount of continual training, builds a set of smaller LLMs. Then they are used as experts, and a router network is trained to route requests to them. Note that the routing decision is one per query and not per token or per layer, and hence the proposal is very different from regular MoE. Empirical results are presented with Llama-7B as the starting point and show that the resulting ensemble has good accuracy-latency tradeoff.

**Strengths:**

Please see the summary. The results seem sound.

**Weaknesses:**

The main results of Table 1 does not support that the proposed method is at a better accuracy vs inference-cost trade-off. The average accuracy of the proposed method is only marginally better than Sheared-Llama yet its effective inference size of 4.7B is much larger than 2.7B. In the other direction and comparing against the Llama-7B starting point, the overall accuracy cost of pruning and ensemble is quite substantial from 61.4% to 55.5%.

In additional to the tasks in Table 1, it would help to add perplexity comparison.

**Questions:**

Please see above.

**Limitations:**

Yes

---

> ### Author Rebuttal · Authors · 2024-08-07
>
> Thanks for your time and effort for the review.  We answered your questions as follows.
>
> **[W1 - Effectiveness of Read-ME]** Thank you for your question.
> - We would like to first clarify that our method significantly outperforms Sheared-Llama. For example, Sheared-Llama nearly performs at random guess levels on the MMLU task (achieving only 26.4% accuracy on a 4-class classification), whereas Read-ME achieves a much higher accuracy of 38.9%.
> - Second, our method is also more training-efficient, using only 1 billion tokens compared to Sheared-Llama's 50 billion tokens (*x50* times higher than ours). To further back up our claim, we perform the Sheared-llama continued tuning using the same amount of tokens as ours. We show the downstream task performance in Table A. By using 1B tokens for tuning, the performance of Sheared-llama reduces to 48.2%, which is significantly lower than our method (55.5%). Note that we use the publicly released Sheared-llama pruning checkpoint [1] as the initialization.
> - Additionally, our method aims to convert a dense pre-trained model into a MoE for efficient inference, and such a compression method is typically not lossless. However, our method achieves a superior accuracy-cost trade-off compared to all baseline compression methods. To validate our approach, we plot the MMLU performance of our method alongside a large number of baseline methods in Figure 4. This data is also presented in tabular form in Table B. Our method outperforms all baseline models/methods of similar sizes.
>
> Table A: Our method is more training-efficient, using only 1 billion tokens compared to Sheared-Llama's 50 billion tokens (x50 times higher than ours). If running sheared-llama with only 1 billion tokens, the performance drops significantly.
> | Method | Cost | MMLU   | Hell. | Wino. | ARC-E | ARC-C | LogiQA | CoQA  | avg.  |
> |:---|:---:|:---:|:---:|:---:|:---:|:---:|:---:|:---:|:---:|
> | Sheared-Llama  | 50B  | 26.4%  | 70.8%  | 67.0% | 67.0% | 41.2% | 28.3%  | 71.7% | 53.2% |
> | Sheared-Llama-efficient  | 1B  | 25.4%  | 59.4%  | 61.9% | 65.8% | 35.8% | 25.1%  | 64.2% | 48.2% |
> | Read-ME  | 1B | 38.9% | 68.5% | 67.7% | 66.6% | 42.3% | 29.7% | 74.8% | 55.5% |
>
> Table B: Evaluation of Read-ME on MMLU benchmark, compared with other representative open-source models and compression techniques.
> | Method | OpenMoE | Sheared-Llama | Pythia | Open-Llama-v2 | LLM-Pruner | Compresso | **Read-ME** | LaCo | SliceGPT | Pythia | Open-Llama-v2 | Llama-2 |
> |:---:|:---:|:---:|:---:|:---:|:---:|:---:|:---:|:---:|:---:|:---:|:---:|:---:|
> | #Param | 2.1B | 2.7B | 2.8B | 3.4B | 4.5B | 4.5B | **4.7B** | 4.7B | 4.8B | 6.9B | 6.9B | 6.9B |
> | MMLU | 26.2% | 26.4% | 26.9% | 25.7% | 23.9% | 25.9% | **38.9%** | 26.5% | 28.9% | 25.5% | 40.2% | 45.3%|
>
>
> [1] https://huggingface.co/princeton-nlp/Sheared-LLaMA-2.7B-Pruned
>
>
>
> **[W2 - Perplexity Comparison]**
> Thank you for your suggestion. We tested the perplexity on Wikipedia, with the results presented in Table C. We set the sequence length to 1024, which is within the sequence limit of all models.
> Table C demonstrates that our method shows negligible performance degradation (from 3.91 to 3.94) compared to Llama-2, the pre-trained dense model.
> Furthermore, our method achieves better performance compared to most of the baseline methods of similar sizes, including Sheared-Llama, and Pythia. We will add the column to Table 1 in the future version, and we have also provided a tentative Table 1 in the uploaded PDF.
>
> Table C: Comparison on Wikipedia perplexity.
> | Model | # Params | Wikipedia PPL |
> |:---:|:---:|:---:|
> | Sheared-Llama | 2.7B | 6.77 |
> | Pythia | 2.8B | 6.02 |
> | Open-Llama-v2  | 3.4B | 3.69 |
> | Read-ME  | 4.7B  | 3.94 |
> | Pythia | 6.9B | 5.49 |
> | Open-Llama-v2 | 6.9B | 2.85 |
> | Llama-2 | 6.9B | 3.91 |

---

> > ### Comment · Reviewer_AX77 · 2024-08-12
> > **Thanks for the rebuttal**
> >
> > I thank the authors for the discussion and has raised my rating from 3 to 4.

---

> > > ### Author Response · Authors · 2024-08-12
> > >
> > > Thank you very much for your thorough review and for the thoughtful comments you provided on our work.
> > >
> > > We have made the necessary revisions in response to your comments, and we hope that our revisions meet your expectations.
> > > If there are any remaining concerns or if you have any further suggestions that could help us improve the quality of our work,  please do not hesitate to let us know. We would greatly appreciate any further guidance you can provide.
> > >
> > > Once again, we are truly grateful for your support.
> > > Authors

---

### Official Review · Reviewer_EGvf · 2024-07-08

**Soundness:** 3
**Presentation:** 3
**Contribution:** 3
**Rating:** 7
**Confidence:** 4

**Summary:**

The paper proposes a novel framework to enhance the efficiency of pre-trained LLMs by transforming them into post-hoc Mixture-of-Experts (MoE) models. The key innovation lies in decoupling the router from the MoE backbone, which facilitates pre-gating and lookahead scheduling, thereby improving memory management and batching during inference. The proposed method, Read-ME, demonstrates significant improvements in both latency and task performance compared to existing models.

**Strengths:**

1. The paper nicely bridges the gap between algorithmic advancements and system-level optimizations. By addressing both fronts, the proposed framework ensures that the improvements in model architecture translate into real-world performance gains.
2. To the best of my knowledge, the introduction of a pre-gating, shared router decoupled from the MoE backbone is a significant innovation. Although breaking down dense LLMs into MoEs alone has been done before, this new gating approach allows for pre-computation and lookahead scheduling, addressing inefficiencies in traditional layer-wise routing systems.
3. The expert-aware batching and optimal expert caching algorithms are well-designed to leverage the pre-gating architecture, showing clear improvements in mean and tail latency.
4. The paper provides comprehensive experimental results that validate the effectiveness of the proposed approach. The improvements in MMLU performance and latency reductions are significant and clearly presented. The comparison with various baselines, including both dense and MoE models, is thorough and demonstrates the superiority of Read-ME across different metrics.

**Weaknesses:**

1. The experimental subject is limited. While the paper demonstrates impressive results on the LLaMA-2 model, the scalability and generalization of the proposed method to other model types and larger scales are not extensively explored. It would be essential to include experiments on more diverse datasets and larger models.
2. Although the paper claims minimal overhead for the auto-regressive router, the detailed analysis of its computational costs relative to traditional routers could be expanded. Can you provide a more detailed breakdown of the computational overhead introduced by the auto-regressive router compared to traditional routers? How does this impact overall inference latency, especially in high-throughput scenarios?
3. The paper could benefit from a more detailed explanation of the pre-gating and batching algorithms, potentially with pseudocode or flow diagrams to aid reproducibility. Sometimes I cannot fully follow the authors’ texts.

**Questions:**

See weakness. Overall, this seems to be reasonably solid LLM co-design work, but my main concern is its limited evaluation to Llama2 only. Reporting addition experiments on Gemma or Mistral would be strongly encouraged.

**Limitations:**

Limitations are adequately addressed.

---

> ### Author Rebuttal · Authors · 2024-08-07
>
> **[W1 - Additional experimental results]** Thanks for the suggestion! To validate that our method remains effective in other scenarios, we use the Mistral model as the pre-trained dense model, and convert it to the MoE structure, with the proposed method. The task is challenging because we do not have prior knowledge on the Mistral original training data, and our experiment in Table A shows that our method remains effective without the prior knowledge of the original training domain.
>
> Table A: Ablation study on Mistral[1] pre-trained model.
> |Method | Pre. domain | FT. Domain | # Param | MMLU | Hell. | Wino. |ARC-E | ARC-C | LogiQA | CoQA | avg. |
> |-|-|-|-|-|-|-|-|-|-|-|-|
> | ReadME-mistral | N/A | Red-Pajama | 4.7B - (17B) | 39.2% | 79.1% | 68.2% | 77.1% | 49.3% | 30.9% | 76.2% | 60.0% |
> | mistral | N/A | - | 6.9B | 62.1% | 84.5% | 79.3% |82.7% | 63.7% |33.5% | 80.3%| 69.4%|
> | ReadME-Llama-2| Red-Pajama | Red-Pajama | 4.7B - (17B) | 38.9% | 68.5% | 67.7% |66.6% |42.3% | 29.7%| 74.8% | 55.5%|
> | Llama-2 | Red-Pajama | - | 6.9B | 45.3% | 78.6% | 69.3% | 76.4% | 53.0% | 31.0% | 75.9% | 61.4%|
>
>
> [1] Mistral 7b.
>
>
> **[W2 - Computational cost of Auto-regressive Routers]**
>
> Thanks for the suggestion!
>
> For a detailed analysis, we added: (1) FLOPs comparison, (2) latency, and (3) latency breakdown with a larger batch size (high-throughput scenarios) of a Traditional Router (TR) and an Autoregressive Router (AR). To focus solely on the router’s impact on latency, we controlled other variables (e.g., the number of activated parameters) to be the same.
>
>
> (1) flops comparison
>
> Table B. Flops comparison between Traditional router and Auto-regressive router
>
> |              | Traditional Router | Auto-regressive Router |
> |-|-|-|
> | flops/sample | 4.7 KFLOPs         | 3 KFLOPs               |
>
>
> (2) latency [ms]
>
> Table C. Latency comparison between Traditional router and Auto-regressive router
>
> |            | bsz=5 | bsz=5 | bsz=10 | bsz=10 | bsz=20 | bsz=20 | bsz=30 | bsz=30 |
> |-|-|-|-|-|-|-|-|-|
> |            | TR    | AR    | TR     | AR     | TR     | AR     | TR     | AR     |
> | Router     |  1.76 |  0.61 |   1.80 |   0.61 |   1.78 |   0.61 |   1.93 |   0.61 |
> | Attention  | 18.13 | 18.18 |  18.28 |  18.13 |  18.49 |  18.36 |  19.59 |  19.66 |
> | Expert/MLP | 22.43 | 21.75 |  24.59 |  22.53 |  24.97 |  22.99 |  30.17 |  28.31 |
> | Sum        | 42.31 | 40.55 |  44.67 |  41.27 |  45.23 |  41.96 |  51.69 |  48.59 |
>
>
> (3) latency breakdown [%]
>
> Table D. Latency breakdown comparison between Traditional router and Auto-regressive router
>
> |            | bsz=5   | bsz=5   | bsz=10  | bsz=10  | bsz=20  | bsz=20  | bsz=30  | bsz=30  |
> |-|-|-|-|-|-|-|-|-|
> |            | TR      | AR      | TR      | AR      | TR      | AR      | TR      | AR      |
> | Router     |   4.15% |   1.50% |   4.02% |   1.48% |   3.93% |   1.46% |   3.74% |   1.26% |
> | Attention  |  42.85% |  44.85% |  40.92% |  43.92% |  40.87% |  43.75% |  37.90% |  40.47% |
> | Expert/MLP |  53.01% |  53.65% |  55.06% |  54.59% |  55.20% |  54.80% |  58.36% |  58.27% |
> | Sum        | 100.00% | 100.00% | 100.00% | 100.00% | 100.00% | 100.00% | 100.00% | 100.00% |
>
>
>
> Note that the computational cost of both the traditional router and the autoregressive router is *theoretically linear to batch size*. Therefore, when the batch size is high (in high-throughput scenarios), the cost increases linearly. In both cases, the computation can be parallelized, so this remains negligible in end-to-end latency even in high-throughput scenarios.
> In fact, we would like to clarify that *the bottleneck in high-throughput scenarios is actually the expert layers*, as seen in table D – Expert/MLP row. This issue can be addressed by the methods discussed in Section 4. *Traditional layerwise routers do not allow for efficient system design*, which underscores the need for a careful co-design of routers.
>
> **[W3 - Reproducibility and Pseudocode]**
>
> In Appendix A, we provide the pseudocode for the batching algorithm and pre-gating. In summary, at each scheduling step, we find the expert with the most requests and select that expert for the current step. We then check whether the scheduled tokens exceed the maximum token length or the maximum number of requests that can fit. This process is repeated until no more requests can be scheduled. We will release the code publicly and refine the text to improve understanding as well.

---

> > ### Comment · Reviewer_EGvf · 2024-08-11
> > **Response to Authors**
> >
> > I thank the authors for their rebuttal and supplemented experiments. I appreciate the reply, and I found my concerns adequately addressed. I will raise the score to 7 in response to the authors' rebuttal.

---

> > > ### Author Response · Authors · 2024-08-11
> > > **Many thanks for raising the score**
> > >
> > > Thank you very much for your insightful suggestions, which have been greatly enlightening and are crucial for enhancing the quality of our paper! We will adhere to these suggestions in the final version and also revise the paper according to all other comments.

---

### Official Review · Reviewer_ZdPD · 2024-07-12

**Soundness:** 3
**Presentation:** 4
**Contribution:** 3
**Rating:** 7
**Confidence:** 4

**Summary:**

This paper proposes Read-ME, a novel framework for pruning large LLMs into smaller MoEs with minimal training cost. Read-ME separates the gating routers from the critical paths of the inference process and trains an individual expert subnetwork to perform offline pre-gating. With this refactorization of MoEs, the paper further demonstrates the effectiveness of Read-ME by designing optimal expert pre-fetching and caching and expert-aware batching for low-latency and high-throughput serving. Extensive experiments show that Read-ME produces small MoEs that outperform existing small dense models with significantly higher inference performance, lower latency, and higher memory efficiency.

**Strengths:**

- The paper is well-written, well-organized, and easy-to-follow.
- The idea of pruning LLMs into smaller MoEs is quite interesting.
- Experiments are thorough and extensive.

**Weaknesses:**

- Overall, this is a very interesting work. My biggest question is: what's the motivation behind pruning LLMs into smaller MoEs? If I need a smaller model, why don't I just prune an LLM and have a small but dense model, if the small dense model has approximately the same (or even less) number of activated parameters?

- The observation of Figure 2 is interesting. However, what's the dataset (or input tokens) you use for plotting this figure? Does this observation still hold if the dataset (or input tokens) changes significantly?

- Section 2.3, "The above observations suggest that among many routing paths, only a few of them are in use during the inference..." with only mutual information between adjacent layers may not justify this assumption. It would be better to visualize the end-to-end routing paths (from the first to the last layer) and explicitly show that only a few paths are used frequently.

- The idea of separating routing logic from the inference process is not new. For example, [1] tries to distill the knowledge of gating networks and performs offline routing. How do you compare your expert subnetwork with [1]? Does training the expert subnetwork cost more than KD-based methods?

- To follow up on the previous comment, the evaluation does not compare Read-ME with any existing offline routing approaches like [1]. If the major benefits of Read-ME come from separating routing logics, then the paper's contributions would decrease.

- In evaluation, comparing Read-ME 4.7B with those baselines with fewer parameters may be unfair. Since Read-ME is pruned from much larger LLMs and still holds more activated parameters than baselines, it wouldn't be surprising to see that Read-ME has the best performance. This performance increase comes at the price of requiring more GPU memory. Once again, this question goes back to the motivation: why would someone need to prune LLMs into smaller MoEs instead of smaller dense models? Comparing Read-ME MoEs with smaller dense models pruned from the same LLM may help address this question.

- This pruning process may need a theoretical justification on the performance guarantee, i.e., smaller MoEs are guaranteed to not suffer large performance degradations.


[1] SiDA: Sparsity-Inspired Data-Aware Serving for Efficient and Scalable Large Mixture-of-Experts Model. MLSys'24

Minor issues:

- Figure 2(c) is never mentioned.

- Figure 3 does not have indices for sub-figures.

**Questions:**

Pleease refer to the Weaknesses.

**Limitations:**

Pleease refer to the Weaknesses.

---

> ### Author Rebuttal · Authors · 2024-08-07
>
> Thanks for all the interesting questions. Please see below.
>
> **[W1 - MoEs Motivation]** We validated that MoE achieves better cost-accuracy trade-off than small dense models acquired by pruning, and provided the results in Appendix C.1.
>
> We mentioned that prior compression efforts focus on converting large dense pre-trained models into smaller dense models. However, we argue that a smaller MoE model, with fewer activation parameters, is a better target architecture. To ensure a fair comparison, we (1) create a 4.7B parameter-dense model matching a single expert network size, and (2) fine-tune it for the same number of steps.
> Table 5 shows refactorizing the pre-trained model into an MoE structure, rather than a smaller dense variant, leads to significant performance improvement.
>
> Table 5: By adopting an MoE as the target structure instead of a dense model, our model achieves better overall performance.
> |Eval|Arxiv|Books|C4|CC|Github|Stack.|Wiki.|MMLU|
> |-|-|-|-|-|-|-|-|-|
> |Dense|5.63|1.94|11.78|9.68|3.75|13.42|6.24|27.1%|
> |Read-ME|4.18|1.31|10.57|7.72|2.39|12.52|3.94|38.9%|
>
> **[W2 - Figure 2 Details]** For Figure 2, we used the Arxiv dataset, a subset of Red-pajama. The observation holds with Wikipedia and Github subsets as well. Please see the uploaded PDF for visualization results; we will add these in a future version.
>
> The visualized high mutual information between two adjacent layers’ expert selection is sufficient to support our claim. Since $H(S_1, …, S_L) \le \sum_l H(S_l) - \sum_l I(S_{l+1}; S_l)$, high layer-wise mutual information implies low-entropy (deterministic) path selection.
>
> Regarding visualizing end-to-end routing paths, firstly we would like to mention that since the model has 32 layers and each layer performs Top-2 selection out of 8 experts, there will be ${8 \choose 2}^{32}$ (approximately $2 \times 10^{46}$) possible paths.
>
> Instead, we calculated routing statistics for 180k tokens, finding only 8.4k paths out of $2\times 10^{46}$ possible paths activated at least once. The top 20 paths are selected by 2805 tokens, and the top 40 paths by 4947 tokens. The observation validates that only a small fraction of routing paths are frequently selected. Please see the uploaded PDF for more visualization.
>
> **[W3 - Comparison with SiDA]** SiDA is an alternative approach that separates the router from the inference path. However, Read-ME offers better opportunities to improve inference efficiency because it is designed to enable ***expert-aware batching.*** Additionally, we would like to emphasize that, by design, the Read-ME router provides ***exact expert selection,*** while SiDA and prior works offer approximate selection [1,2]. In detail,
>
> (1) Batching and latency:
> With SiDA, *tokens cannot be batched together* because they do not share routing decisions across all layers, unlike Read-ME (Since SiDA is distilled from the Switch Transformer, it is not possible to change decisions of routers). This vastly magnifies the expert space when  batching and makes it difficult to compose an efficient batch that is aware of expert selection.
> In effect, SiDA activates more experts for each batch at each layer, leading to increased latency. Table A compares the inference latency between SiDA and Read-ME. (SiDA did not release a checkpoint, so we used SwitchTransformer-8, the teacher model from which SiDA was distilled, as a proxy)
> Table A. Inference latency of SiDA and Read-ME.
> | |SiDA|Read-ME|
> |-|-|-|
> |Latency[ms]|62.34|48.59|
> |Avg # of Activated Experts|5.51|3.51|
>
> (2) Router accuracy:
> SiDA’s offline routing function is an approximation method that is distilled from original layerwise routers. Thus, the accuracy of routing is not 100%. Table B shows the “failure rate” of the SiDA router (defined as prediction miss rate on the expert activation of the trained router), and Table C shows the resulting degradation in task performance. With an incorrect guess, the entire expert selection can fail, leading to a collapse in inference, especially as the model scales. In contrast, *our method is exact, ensuring no performance drop during inference*, regardless of model size.
>
> Table B. SiDA Router failure rate
> |#experts|SST2|MRPC|MultiRC|
> |-|-|-|-|
> |8|-1.00%|-2.59%|-8.26%|
> |128|-1.22%|-1.35%|-9.51%|
>
> Table C. Accuracy drop due to SiDA’s Router Failure
> |#experts|SST2|MRPC|MultiRC|
> |-|-|-|-|
> |8|-1.75%|-2.51%|-1.05%|
> |128|-6.98%|-7.41%|-7.44%|
>
> (3) Training cost:
> Note that SiDA is based on an MoE model and only distills the router, whereas our model is based on a dense model and builds both the expert network and the router, training them together. This means that SiDA's training time only accounts for the router training cost, while our training time includes both the router training cost and the expert specialization cost. Additionally, SiDA did not report the training cost in terms of tokens, making a meaningful comparison of training costs impossible.
>
> In addition, our work introduces novel contributions to caching and prefetching, which SiDA lacks. SiDA relies on on-demand expert loading and FIFO-based expert eviction, which can negatively impact performance. Please refer to Section 4 for a detailed discussion of our contributions in these aspects. We will add discussion of SiDA in the revised version.
>
> [1] Fast inference of mixture-of-experts language models with offloading
>
> [2] SiDA: Sparsity-Inspired Data-Aware Serving for Efficient and Scalable Large Mixture-of-Experts Model
>
> **[W4 - Theoretical Analysis]** In this work, we empirically examine the redundancy of layer-wise routers and propose a system-oriented method to convert a pre-trained dense model to an MoE with minimal additional training costs. While our focus is on empirical validation, providing a theoretical justification is beyond the scope of our paper and something we hope to explore in future study.
>
> Thanks for catching the missing references and indices. We will definitely correct it in our revised version.

---

> > ### Comment · Reviewer_ZdPD · 2024-08-12
> >
> > I appreciate the thorough answers from the authors. The rebuttal has addressed all my questions, provided interesting new results, and revealed the novelty of this paper.
> >
> > In response to the rebuttal, I would like to raise my rating to help advance the possibility of acceptance for this paper.

---

> > > ### Author Response · Authors · 2024-08-12
> > >
> > > Many thanks for your thoughtful comments and suggestions. We value your support and will be incorporating all your feedback in our revised version

---

### Official Review · Reviewer_wHVe · 2024-07-15

**Soundness:** 3
**Presentation:** 3
**Contribution:** 3
**Rating:** 6
**Confidence:** 3

**Summary:**

The paper proposes an inference-aware method to convert a pretrained dense model into a Mixture-of-Experts architecture, where each expert is smaller than the original dense model. To extract the different experts, they use a dataset from given subdomain to identify the top activated channels. To route among the experts, the authors first break down the limitations of current per-layer routing schemes (waiting for all tokens at the layer to finish, redundancy across layers). They propose to decouple the router from the base model, and instead train a 1-layer transformer block as the router, which predicts token expert assignment autoregressively.

Overall, these changes allow for inference friendly deployment, by enabling careful batching of examples that share similar experts, and with better caching of experts in resource constrained settings. The authors show that their approach strikes a good trade-off between efficiency and performance.

**Strengths:**

1. The authors do a good job at breaking down the current bottlenecks in deploying MoE models
2.  The proposed external routing approach is simple and effective
3.  The latency evaluation and experimentation is well designed and clear

**Weaknesses:**

1.  I am a bit surprised that the gains in latency compared to the seed llama models are somewhat small (19%) given that the number of active parameters is reduced by 30%, and that there is also a decent performance drop from the seed llama model. For resource constrained settings, how would the approach of loading llama layer-by-layer fair against Read-ME ? Like Read-ME, this approach can know in advance what layers to load, and could hopefully retain the performance of the full lama model.
2. Some parts of the analysis require further clarification (please see questions)

**Questions:**

1. If you keep the top activated neurons, should equation 3 be an argmax ?
2. What is the impact of the routing distillation loss ? Can you provide an ablation experiment where it is not used to access its impact ?
3. How would figure 3 look with READ-ME instead of standard MoEs ? This would be a good visualization to compare the methods.
4. How were the baseline numbers in Figure 4 obtained ? Did the authors rerun the baselines ? Can you confirm that all these baselines start from the same base model ?
5. The "cost" column of table 1 is misleading; given that your model starts from llama-2, you must either include the compute to create llama-2 in your analysis, or only monitor the additional compute starting from llama 2 (in which case the cost for llama 2 would be 0).

---

> ### Author Rebuttal · Authors · 2024-08-07
>
> **[W1 - Comparison with layer-by-layer loaded Llama ]**
> Thanks for the question. We compared the latency of the layer-by-layer loaded llama and Read-ME model in the following table.
>
> Table A. Latency comparison
>
> |Method| Latency [ms] |
> |---------|--------------|
> | Layer-by-layer Llama-7b |     111.909 |
> | Read-ME |      91.531 |
>
> The difference arises from the size of the parameters to be loaded. The size of Read-ME's expert layer (including both the top-1 expert and residual expert) is approximately 25.0% that of the MLP counterpart in the LLaMA baseline. Regarding peak memory usage, Read-ME exhibits a 10% lower profile, though both models consume an insignificant amount of memory overall.
>
> Our method converts a dense pre-trained model into a MoE for efficient inference. Although this compression is generally not lossless, please note that our approach offers a better accuracy-cost trade-off than all baseline compression methods.
>
> **[Q1 - Understanding of Equation 3]** Sorry for the confusion. We found this is a typo in our original submission., As we are selecting the mask $\boldsymbol{M}$ to maximize the magnitude of activated channels, the operator should indeed be $\arg\max$ in Equation 3.
>
> **[Q2 - Ablation on Routing distillation loss]** Thank you for the suggestion. We have ablated the routing distillation loss by removing the router training step and using a random routing mechanism instead, tuning only the expert weights using the language modeling loss. To ensure a fair comparison, we maintained the same number of training tokens and the same training schedule. The resultant performance is provided in Table B. The results show that with the routing distillation loss, the average downstream task accuracy increased from 51.5% to 55.5%, validating the necessity of the routing distillation loss.
>
> Table B: Ablation study on routing distillation loss. We compare the performance with and without the routing distillation loss, while keeping the number of training tokens, to validate the necessity of routing distillation loss.
> | Method | MMLU | Hell. | Wino.  | ARC-E | ARC-C | LogiQA | CoQA | avg.|
> |:---:|:---:|:---:|:---:|:---:|:---:|:---:|:---:|:---:|
> | Read-ME (w/ routing distillation loss)| 38.9% | 68.5%  | 67.7%  | 66.6% | 42.3% | 29.7% | 74.8% |  55.5% |
> | Read-ME (wo routing distillation loss)| 30.6% | 65.1%  | 65.8%  | 64.2% | 39.9% | 24.9% | 69.7% |  51.5% |
>
> **[Q3 - Read-ME compared to Figure 3]**
> This is a great suggestion! Please check Fig 3 of the PDF uploaded.
> In Fig 3-left, Read-ME batches tokens directed to the same expert, so all arrows point to a single expert at each layer. Also, there is only one router instead of three, resulting in all arrows following the same path from layer 1 to layer 3, leading to a unique activated expert count of 1.
> In Fig 3-middle, Read-ME shifts the distribution to the lower left, where most points stay within the range of x < 4.5 and y < 55.
> Fig 3-right already compares the traditional approach with Read-ME.
>
> **[Q4 - Baseline numbers in Figure 4]** Thanks for asking.
> - MMLU is the common evaluation benchmark for LLMs, so most of the numbers in Figure 4 are collected from their original paper, including OpenMoE, Llama-2, LaCo, and Compresso; For Sheared-Llama, Open-Llama and Pythia, as they do not report MMLU in their original paper, we test their publicly released checkpoints with the lm-eval-harness library[1]; LLM-Pruner and SliceGPT didn’t report MMLU nor release checkpoints, so we use the numbers reported in LaCo [3].
> - Yes, all of the methods reported in Figure 4 use the same base model - Llama-2.
>
> **[Q5 - Training Cost Calculation]** Thanks for the suggestion. First, our training cost calculation follows Sheared-Llama[2] (see Table 1 of the paper), a representative post-training method to generate a small model out of a large one by pruning. Second, the training cost here measures the computational resources needed by the LLM deployer to obtain a new model of an auxiliary architecture (a smaller model in the Sheared-Llama case, and a MoE in our case), given all the publicly available resources. Using the publicly released checkpoints (e.g. Llama-2) will not incur additional training costs. We appreciate the suggestion and will mention this in the caption. Please see the PDF for our tentative Table 1.
>
> [1] https://github.com/EleutherAI/lm-evaluation-harness
>
> [2] Sheared LLaMA: Accelerating Language Model Pre-training via Structured Pruning
>
> [3] LaCo: Large Language Model Pruning via Layer Collapse

---

> > ### Comment · Reviewer_wHVe · 2024-08-12
> > **Re: rebuttal**
> >
> > The authors have provided a very clear rebuttal and detailed answers to my initial set of questions. I will change my score accordingly

---

> > > ### Author Response · Authors · 2024-08-12
> > >
> > > Thank you once again for your insightful comments and suggestions. We greatly appreciate your support of our work and will be incorporating all the feedback into our revised version.

---

### Author Rebuttal · Authors · 2024-08-07

We thank all reviewers [R1(wHVe), R2(ZdPD), R3(EGvf), R4(AX77)] for their thoughtful and constructive feedback. We are grateful that the reviewers found our approach interesting and effective [R1,R2,R3], the paper well-written and well-organized [R2], and the experimental results thorough and extensive [R2,R3].

We have thoroughly addressed all of the concerns raised by the reviewers. As a model compression effort, we emphasize our superior accuracy-cost tradeoff and training efficiency compared to all baseline methods. We provided additional experiments on routing distillation loss (R1), dense counterpart comparisons (R1,R2, R4), other evaluation metrics (R4), cost analysis of the auto-regressive router (R3), and generalizability to other model families (R3).

For other questions, we added detailed discussions for each reviewer. Please check our PDF if it is mentioned in the answer.

---

### Decision · Program_Chairs · 2024-09-25

**Decision:**

Accept (poster)

**Comment:**

The manuscript has been reviewed by four reviewers. After the rebuttal, three reviewers gave positive ratings, while one gave borderline rejection.

Overall, the reviewers praised the submission for the motivation, the presentation, and experimental validation.

The AC agrees with consensus from the reviewers, and believes that the manuscript, despite not flawless, should be presented to a large audience.

Please, however, do account for the comments in the final version.

Congrats!